# Robust Multi-modal Learning with Shifted Feature Reweighting Against Spurious Correlations

## Abstract

Pre-trained multi-modal models have recently garnered significant attention due to their adaptability to diverse downstream tasks via fine-tuning. However, their resilience to certain group shift issues, *i.e.*, spurious correlations, remains imperative yet relatively under-investigated. We study this problem in vision-language models (VLMs), and we observe potential vulnerabilities in pre-trained VLMs, such as CLIP, when confronted with spurious correlations. While recent studies have been exploited to address unimodal group-imbalances by minority group up-sampling or creating group-balanced subsets, we posit that true robustness can be achieved by debiasing the training process through feature reweighting. In this paper, we propose **S**hifted **F**eature **R**eweighting (**SFR**), a robust multi-modal learning method to mitigate the reliance on spurious features. Specifically, we introduce a novel disagreement-based importance weight that allocates distinct weights to individual instances within the training data. This contrasts with existing group rebalance weight strategies, which uniformly weigh all instances within a group. Our reweighting strategy adeptly addresses disparities in instance-level learning difficulty. Moreover, our empirical results unveil that representation collapse may arise during fine-tuning. To address this, we proposed to introduce feature dropout and show that this simple method can further regularize the training on the majority groups and encourage the training on the minority groups. Empirical results on multiple benchmarks verify our claims and confirm the effectiveness of our proposed SFR. Theoretically, we analyze the performance of our SFR and confirm its superiority in mitigating spurious correlations. Our codes will be here.

## 1 Introduction

Large-scale transfer learning has recently become *show-stealer* in modern deep learning. Pre-trained vision-language models (VLMs), such as CLIP (Radford et al., 2021), in particular, are now setting benchmarks by achieving state-of-the-art performance across a spectrum of real-world computer vision (Radford et al., 2021; Jia et al., 2021; Lai et al., 2023a; Li et al., 2021; Wang et al., 2022; Zellers et al., 2021; Bain et al., 2021; Lai et al., 2023c), and natural language processing tasks (Wang et al., 2018; Talmor et al., 2018; Liu et al., 2019; Yang et al., 2019; Wei et al., 2022; Lai et al., 2023b). As model parameter counts have soared from millions to billions in recent years, fine-tuning large pre-trained VLMs for specific downstream tasks has emerged as a promising approach, balancing performance gains with minimal computational requirements.

Amidst this rapid advancement in VLMs, the burgeoning field of model robustness seeks ways to enhance the resilience of these models, aiming to diminish *spurious correlations* – features that are indicative of the target class within the training dataset, yet inconsequential to the true classification function – without compromising performance (Yang et al., 2023a; Zhang et al., 2022b). More specifically, utilizing standard Empirical Risk Minimization (ERM) for training may inadvertently suffer from spurious correlations, potentially undermining robustness, particularly for underrepresented groups (Geirhos et al., 2020). Such an approach risks marginalizing specific minority subsets within the training data, constraining the model's effectiveness in safety-critical scenarios. For example, in Waterbirds (Sagawa et al., 2020), which involves classifying between "waterbird" and "landbird", a noticeable bias exists. Specifically, the "land" and "water" backgrounds

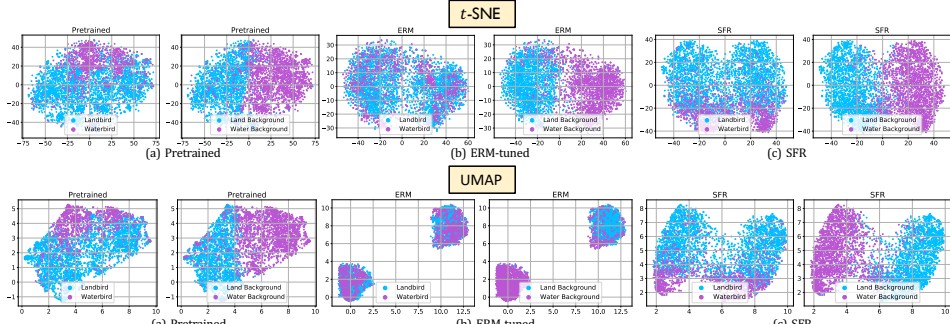

Figure 1: **Representation Visualization for Pretrained CLIP, ERM-tuned CLIP, and our proposed SFR on the Waterbirds Dataset.** The visualizations in the first and second rows use $t$-SNE and UMAP techniques, respectively. These visualizations reveal that both pre-trained and ERM-tuned CLIP rely on spurious features for prediction, characterized by distinct separation patterns that are aligned with spurious attributes, such as the background, instead of the intended target class. In contrast, our SFR method displays considerably improved class separation, highlighting its effectiveness in mitigating spurious correlations and enhancing the robustness of the model.

are spuriously correlated with the "landbird" and "waterbird" classes, respectively. This results in underrepresented groups such as "landbirds on water" and "waterbirds on land".

In the pursuit of improving **group robustness** of *vision* models (Sagawa et al., 2020; Nam et al., 2022; Liu et al., 2021; Creager et al., 2021; Nam et al., 2020; Izmailov et al., 2022; Kirichenko et al., 2023), many existing works underscore the importance of mitigating *spurious correlations* as a pivotal strategy for improved model performance. Yet, a largely uncharted area remains: assessing the robustness of these VLMs against group shifts. Prior work (Yang et al., 2023a) has shown the promise of enhancing VLMs' group robustness via efficient fine-tuning. However, the pursuit of advancing accuracy across diverse groups, often at considerable computational costs (*i.e.*, fine-tuning entire models), remains a challenge that is yet to be well-addressed. In this work, we ask: *How to improve group robustness in pre-trained VLMs with the minimum computational costs?*

In this work, we advance state-of-the-art group robustness of VLMs for image classification. We first explore the feasibility of improving group robustness for popular large pre-trained model, such as CLIP (Radford et al., 2021), constrained by group imbalance. One straightforward approach is to upsample the minority group (Sagawa et al., 2020) or create a group-balanced dataset (Kirichenko et al., 2023; Izmailov et al., 2022). However, our findings (*e.g.*, Figures 1 and 3) suggest these methods may overlook variations in the difficulty of training samples, presumably due to the uniform weighting of all samples within a group. The relative importance of individual samples within each group remains an open question. This encourages us to start attending nonchalantly at trivial samples and become scrupulous at non-trivial samples. In addition to the above issue, recently several works (Andonian et al., 2022; Yang et al., 2022) investigate the intriguing phenomenon that the current contrastive learning paradigms tend to neglect hidden semantics inside the textual description, prominently due to overfitting of the learned visual representations to the certain textual features during training. This implies a trend toward *representation collapse*, which in turn leads to suboptimal performance.

In this paper, we propose **S**hifted **F**eature **R**eweighting (**SFR**), a robust multi-modal learning method to mitigate the reliance on spurious features. Our simple, yet highly effective strategy offers several compelling advantages for fine-tuning pre-trained VLMs, specifically: (1) We introduce a novel disagreement-based importance weighting strategy that allocates distinct weights to individual samples within the training data (Sec. 4.1). The reweighting is achieved by a disagreement-based score derived from contrasting an auxiliary ERM model and our target model, using the entropy differential between their predictions. Our empirical findings suggest that such a strategy enables the model to prioritize high-caliber samples, thereby refining prediction accuracy. Importantly, such a reweighting strategy is plug-and-play and can be applied across diverse models beyond VLMs. (2) We further describe a simple training approach to mitigate *spurious correlations* without *representational collapse* (Sec. 4.3). More specifically, we mask random pixels from attention-pooled representations of the images with only dropout (Srivastava et al., 2014). We find that masking a

certain proportion of the attention-pooled representations, e.g., 10%, plays a nontrivial and meaningful implicit regularization role similar to dropout, making the VLMs robust to representation collapse (Sec. 5.2). This simple use of dropout achieves a substantial improvement compared to prior methods. (3) We apply SFR to fine-tune VLMs and show that, empirically, SFR can substantially enhances the model's robustness on multiple benchmarks including Waterbirds (Wah et al., 2011) and CelebA (Liu et al., 2015) (Sec. 5.1). Our theoretical analysis confirms the superiority of SFR in mitigating spurious correlations (Appendix E). Overall, our contributions are:

- We propose a new plug-and-play training framework (**SFR**), to enable fine-tuning the VLMs to mitigate spurious features without collapse. SFR introduces a disagreement-based score that adeptly determines the relative significance of samples, thereby refining the weight allocation in the loss function.

- We present a sample strategy – randomly masking a certain proportion of the attention-pooled representations – to mitigate *representational collapse* and effectively improve group robustness to produces superior performance of pre-trained VLMs.

- Our extensive experiments using CLIP on multiple benchmarks validate the efficacy of our proposed SFR in mitigating spurious correlations. Our in-depth analysis of reweighting behaviors and representation properties further demonstrates improved performance and consistent group robustness. Theoretical analysis of SFR demonstrates the effectiveness of SFR in mitigating spurious correlations in Appendix E.

## 2 RELATED WORK

**Improving group robustness.** In real-world applications, spurious correlations are prevalent. The spurious correlation issue refers to neural networks often prioritize features over shortcuts — shallow features spuriously linked with classification targets, which can pose challenges, especially in high-stakes and safety-critical scenarios (Geirhos et al., 2020; Yang et al., 2023b; Sagawa et al., 2020; Sohoni et al., 2020; Nam et al., 2020; Yaghoobzadeh et al., 2019; Creager et al., 2021; Kim et al., 2021; Asgari et al., 2022; Zhang et al., 2022b; LaBonte et al., 2023; Idrissi et al., 2022; Qiu et al., 2023; Pezeshki et al., 2021). Consequently, there's a burgeoning research focus on elucidating and mitigating their impact on model efficacy. On one hand, computer vision models frequently rely on semantically unrelated attributes, including an image's background (Sagawa et al., 2020; Lai et al., 2023a; Xiao et al., 2021; Moayeri et al., 2022), texture (Geirhos et al., 2019), secondary objects (Rosenfeld et al., 2018; Shetty et al., 2019; Singla & Feizi, 2022), and other extraneous features (Li et al., 2018; Brendel & Bethge, 2019; Lai et al., 2023c). This becomes particularly profound in critical domains like medical imaging, wherein models might inadvertently prioritize hospital-specific markers (Zech et al., 2018) or incidental indicators (Oakden-Rayner et al., 2020) over genuine symptoms of diseases. On the other hand, large language models (LLMs) demonstrate a tendency to capitalize on superficial characteristics, facilitating their strong performance in benchmarks despite potential gaps in true task comprehension. For example, these models might exploit elementary syntactic patterns or lexical overlaps between sentences to infer their connections (Kaushik & Lipton, 2018; Gururangan et al., 2018; Lai et al., 2023b; Li et al., 2023a; Niven & Kao, 2019; McCoy et al., 2019). In contrast, our goal is to *fine-tune* a more robust pretrained VLMs toward group robustness, rather than training *unimodal* models. The intricacies of spurious correlations in pretrained VLMs remain relatively uncharted.

**Robustness and group annotations.** When provided with group annotations (*i.e.* spurious labels together with class labels), there exists a range of methods that consistently yield improved Worst Group Accuracy (WGA). Many studies have shown that utilizing these group annotations can be pivotal for improving model group robustness. Some of these approaches include the optimization of worst-group loss (Sagawa et al., 2020), cultivating invariant or diverse features (Arjovsky et al., 2019; Goel et al., 2021; Zhang et al., 2022a; Xu et al., 2022), adopting class or group balancing strategies (Cui et al., 2019; Menon et al., 2020; Idrissi et al., 2022; Kirichenko et al., 2023; Izmailov et al., 2022), and employing contrastive learning techniques (Taghanaki et al., 2021; Yang et al., 2023a). Among previous explorations, (Yang et al., 2023a) are the closest to our setting, emphasizing the fine-tuning of multimodal models to mitigate spurious correlations. However, they does not explore importance reweighting strategies, relying solely on uniform weighting across samples within a group, which can hardly bring group robustness boost to real-world applications.

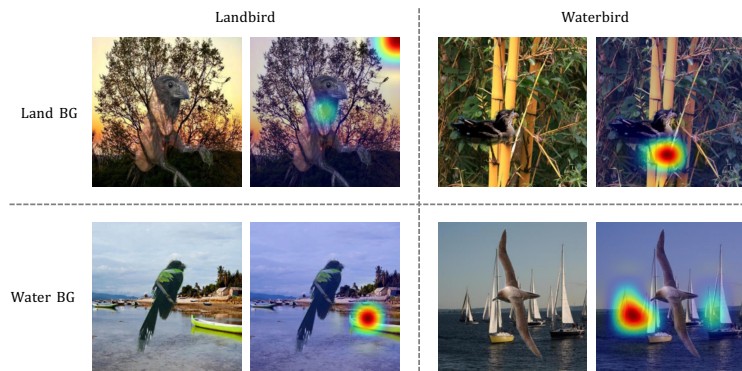

Figure 2: **GradCAM of pre-trained CLIP with ResNet-50 architecture.** For each group of Waterbirds, we show the GradCAM of the visual branch of pre-trained CLIP, which manifests the model's attention towards regions of the images. We observe that on Waterbirds, pre-trained CLIP often attends to the background (BG) of the image rather than the bird itself, indicating the cause of the spurious correlation.

## 3 PRELIMINARIES

**Learning with Spurious Correlation.** Spurious correlation manifests when significant discrepancies in group information exist between the target distribution $\mathcal{P}$ and source distribution $\mathcal{Q}$ from which training samples are drawn, constituting a form of distribution shift (Ming et al., 2022; Zhou et al., 2021). In our study, we focus on classification tasks within the context of group robustness (Sagawa et al., 2020), where the input is denoted as $\mathbf{x} \in \mathcal{X}$ and the its class label as $\mathbf{y} \in \mathcal{Y}$. We posit that the data distribution contains multiple distinct *groups* $g \in \mathcal{G}$, often delineated by amalgamating the class label $\mathbf{y} \in \mathcal{Y}$ and a certain spurious attribute $s \in \mathcal{S}$ where $\mathcal{S}$ is the set of spurious attributes. For example, considering the Waterbirds dataset (Sagawa et al., 2020), the task is to classify $\mathbf{y}$ as either *landbird* or *waterbird*. The background imagery acts as the spurious attribute $s$, with potential values $\mathcal{S} = \{$*land, water*$\}$, leading to groups defined by the combinations of class labels and spurious attributes: $\mathcal{G} = \mathcal{Y} \times \mathcal{S}$.

An attribute $s$ is labeled spurious when it shows correlation with $\mathbf{y}$ but lacks causality. Illustratively, in Waterbirds (Sagawa et al., 2020), nearly 95% of instances tagged as $\mathbf{y} =$ waterbird exhibit the spurious attribute $s =$ water. Such tendencies may drive models to excessively rely on the background (like water) for predictions, undermining accuracy for underrepresented groups such as $g =$ (landbird, water). Following (Kirichenko et al., 2023; Zhang & Ré, 2022; Zhang et al., 2022b; Qiu et al., 2023; Yang et al., 2023a; LaBonte et al., 2023; Izmailov et al., 2022; Yang et al., 2023b), we employ *worst group accuracy* (WGA) for performance evaluation, representing the lowest accuracy achieved across all groups.

**Engaging with Spurious Attributes.** Prior works aiming for robustness against spurious correlations predominantly focuses on training *unimodal* models (Sagawa et al., 2020; Kirichenko et al., 2023; Nam et al., 2022; Sohoni et al., 2021). In contrast, our study delves into *fine-tuning* pre-trained VLMs for enhanced robustness, with a keen emphasis on non-trivial (minority) groups, while maintaining computational efficiency.

**An Overview of CLIP.** In this work, we primarily use Contrastive Language-Image Pre-training (CLIP) (Radford et al., 2021)[1] as our VLM choice due to its state-of-the-art performance. CLIP trains models using over 400M image-caption pairs from the web. The central architecture to CLIP includes the following: (i) a visual encoder, (ii) a text encoder, and (iii) their outputs' dot product, termed the 'alignment score'. Essentially, for a batch of $N$ image-caption pairs, each image should correspond with its given text. For an image $\mathbf{x}_i$, we denote $\mathbf{v}_i$ as its image representation and $\mathbf{u}_i$ as the corresponding text representation. Specifically, the prediction probability of image representation $\mathbf{v_i}$ aligning with corresponding caption representation $\mathbf{u_j}$ is denoted as $\exp(\beta \mathbf{v_i}^T \mathbf{u_j}) / \sum_{k=1}^{N} \exp(\beta \mathbf{v_i}^T \mathbf{u_k})$, where $\beta$ is a tunable parameter.

---

[1] https://github.com/openai/CLIP

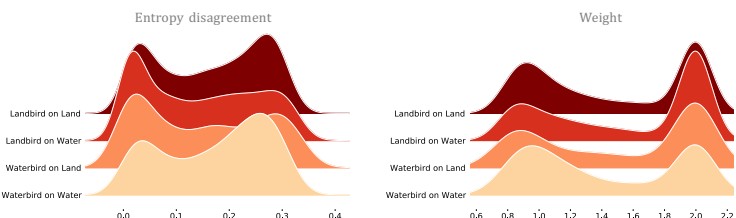

Figure 3: **Sample-wise Distribution of Entropy Disagreement and Weight.** We compare the distribution of entropy disagreement $\delta(\mathbf{x}, \mathbf{y})$ and corresponding weights $\mathrm{w}(\mathbf{x}, \mathbf{y})$ for each group in the Waterbirds dataset, as shown in Eq. equation 1 (left subplot) and Eq. equation 2 (right subplot), respectively. The range has been slightly clipped to enhance clarity in presentation. Notably, the minority groups (*i.e.*, "landbirds on water" and "waterbirds on land") show a higher prevalence of smaller values in entropy disagreement compared to the majority groups. Consequently, the weight distribution for these minority groups showcases a greater proportion of larger values, as per our weight definition, which is inversely related to entropy disagreement.

## 4 SHIFTED FEATURE REWEIGHTING

In this section, we introduce our proposed method, SFR, as a straightforward yet highly effective pipeline aimed at addressing the issue of spurious correlation inherent in pre-trained CLIP, as described in Figure 6 (in Appendix).

**Overview and Motivation.** Our framework focuses on fine-tuning the pre-trained CLIP. That is, we solely train the final layer (*i.e.*, the projection head of the visual branch of CLIP), while keeping the remainder of the model frozen. The rationale behind our approach is grounded in several key factors. Firstly, it is designed to be lightweight and efficient. Furthermore, when visualizing feature attention using GradCAM (See Figure 2), we observe that the pre-trained CLIP model tends to attend spurious image features rather than the core features, such as the background in the Waterbirds dataset. To gain more insight, we also visualize the represetations of pre-trained CLIP, ERM-tuned CLIP, and our proposed SFR on the Waterbirds Dataset. This demonstrate the presence of spurious correlations (See Figure 1). These can be attributed to a well-known phenomenon in neural networks where they tend to prioritize learning easier features, often neglecting the more critical aspects of the data (Geirhos et al., 2020).

In addition, spurious correlations involve a distribution shift where model performance relies on group information in the training data rather than on the true underlying relationship. An intuitive golden solver to address this issue is importance sampling, which assigns varying weights to samples to mitigate group bias(Yang et al., 2024). However, in practical scenarios, directly applying importance sampling either leads to poor performance due to high-variance estimation (Byrd & Lipton, 2019) or incurs significant computational costs, limiting its effectiveness in complex data environments (Bugallo et al., 2017; Robert, 1999). To address these bottlenecks, we introduced a disagreement-based reweighting score that mimics true importance weights $\mathrm{w}(x)$, dynamically adjusting sample weights to efficiently reduce spurious correlations and enhance model reliability across diverse data distributions. Driven by these insights, we here develop a streamlined, lightweight fine-tuning approach specifically designed to counteract spurious correlations.

The performance challenges encountered by ERM in datasets with spurious correlations stem from two primary factors: (1) *geometric skew*: an imbalance in the classification margins between different groups within a dataset, particularly when an ERM-trained classifier, treats majority and minority groups inequally. This results in an equalization of training margins across these groups, potentially leading to biased or less effective learning outcomes, especially for minority groups. For example, as shown in Figure 3, in the Waterbirds dataset, we observe the distribution difference of entropy disagreement $\delta(\mathbf{x}, \mathbf{y})$ and corresponding weights for each group, thereby revealing the gaps in the entropy levels across different groups; and (2) *statistical skew*: the tendency of a neural network to prioritize "easy-to-learn" spurious attributes, rather than focusing on the true label information, particularly in datasets with spurious correlations. This often occurs due to the slow convergence of gradient descent on minority portion of the data. The network might then overly rely on these spurious attributes, especially in early stages of training or in scenarios where training is insufficient.

To this end, our approach includes two major components: disagreement-based reweighting and feature dropout, which are supplemented by specific loss functions that consistently achieve competitive performance. Importantly, it is noteworthy that the versatility of these techniques extends beyond CLIP and is not confined to particular loss functions, making them valuable plug-and-play tools with broad applicability across various domains.

## 4.1 DISAGREEMENT-BASED REWEIGHTING

Our preliminary findings underscore the need for a reweighting strategy tailored to pre-trained foundation models, *e.g.*, CLIP. Specifically, we aim to devise a method for quantifying the difficulty of individual samples, thereby enabling us to assign appropriate weights for more robust training. This concept draws inspiration from significant research in the field (Nam et al., 2020; Sohoni et al., 2020; Creager et al., 2021; Liu et al., 2021; Zhang et al., 2022b; Zhang & Ré, 2022; Duan et al., 2024; LaBonte et al., 2023), often involving an auxiliary model to identify challenging or minority group samples, either implicitly or explicitly. Such methods include pseudo-labeling with ERM model predictions (Nam et al., 2020), subset selection (Liu et al., 2021), cluster-based grouping using ERM features (Sohoni et al., 2020), and contrastive learning (Zhang et al., 2022b; Zhang & Ré, 2022).

Our method, while resonating with these established approaches, diverges in its practical application. We propose a novel importance weighting strategy, assigning distinct importance weights to each sample based on its difficulty level. According to Theorem 3 in (Cortes et al., 2010), the discrepancy between the generalization error on target distribution, $R(h) = \mathbb{E}_{x\sim\mathcal{P}}[\mathcal{L}(h(x), f(x))]$, and the weighted empirical loss on source distribution, $\hat{R}_{\mathrm{w}}(h) = \frac{1}{m}\sum_{i=1}^{m} w(x_i)\mathcal{L}(h(x_i), f(x_i))$, is upper bonded, providing a theoretical guarantee for our design. A more detailed theoretical analysis is in Appendix E.

Despite the theoretical appeal of employing importance weights $\mathrm{w}(x)$ in ERM models, practical implementation encounters significant challenges. These include weight degeneracy (Robert, 1999), time-consuming iteration process to approximate target distribution (Bugallo et al., 2017), and high-variance estimates in high-dimensional data (Byrd & Lipton, 2019). Our proposed design effectively addresses the aforementioned challenges without requiring prior knowledge of distribution or expensive computing iterations, which leads to a lightweight, computationally efficient, plug-and-play algorithm. Further theoretical analysis refer to Appendix E.2.

The implementation of our disagreement-based reweighing strategy involves two primary stages. Initially, we train an ERM model on the downstream dataset such as Waterbirds using the standard Cross-Entropy loss. However, unlike previous methods, this ERM training does not update the entire model. Instead, we optimize the use of the pre-trained CLIP model's robust representation, focusing solely on fine-tuning the vision branch's last-layer projection head. We then determine a score based on the entropy disagreement between this ERM model and our target model.

Specifically, we use the ERM model as an auxiliary model in order to fine-tune a separate target model. The target model is initialized by a pre-trained CLIP. For each sample $(\mathbf{x}, \mathbf{y})$, we define the per-class disagreement score as:

$$\delta(\mathbf{x}, \mathbf{y}) = \left| \mathbb{P}_{\mathrm{ERM}}(\mathbf{y}|\mathbf{x}) \log \frac{1}{\mathbb{P}_{\mathrm{ERM}}(\mathbf{y}|\mathbf{x})} - \mathbb{P}_{\theta}(\mathbf{y}|\mathbf{x}) \log \frac{1}{\mathbb{P}_{\theta}(\mathbf{y}|\mathbf{x})} \right|, \tag{1}$$

where $\mathbb{P}_{\mathrm{ERM}}$ and $\mathbb{P}_{\theta}$ denote the prediction probability of the ERM model and the target model, respectively. This score quantifies the entropy discrepancy between the two models, offering insight into sample difficulty. Without depending on prior information for density selection, our disagreement-based score can be easily adapted to various scenarios with minimal modifications. Our score skillfully avoids the high-variance issues identified in (Byrd & Lipton, 2019) by implicitly calculating the divergence between the source distribution $Q$ and target distribution $P$ using entropy, which effectively approximates the true importance weights. When the ERM model and the target model exhibit significant disagreement in their prediction certainty level for the label $\mathbf{y}$ of a sample $\mathbf{x}$, the score $\delta(\mathbf{x}, \mathbf{y})$ will be large. The final weight for a sample $(\mathbf{x}, \mathbf{y})$ is then defined as the inverse of this score:

$$\mathrm{w}(\mathbf{x}, \mathbf{y}) = 1/\delta(\mathbf{x}, \mathbf{y}). \tag{2}$$

This weight[2] with a hyperparameter $W > 0$. Details in Appendix C is integrated into the training objective (Eq. 5) in our experiments. Our extensive empirical results show that this weight incorporated into CLIP achieves significant performance gain, as will be detailed in Section 5.

## 4.2 INCORPORATION INTO CLIP

In this section, we delve into the practical implementation of the sample reweighting strategy, a flexible approach compatible with various training objectives. To assess its effectiveness, we apply it in conjunction with two loss functions specifically designed to address spurious correlations, both of which have achieved state-of-the-art (SoTA) performance in fine-tuning CLIP.

**Spurious-aware Contrastive Loss (SCL).** Our first exploration focuses on the spurious-aware contrastive loss, a loss function that has been employed successfully in mitigating spurious correlations (Yang et al., 2023a). This loss has achieved SoTA performance in this particular setting. We assume that group information is either readily available or inferred through attribute detection methods like OWL-ViT (Singla et al., 2021; Minderer et al., 2022; Yang et al., 2023a). Considering a single data point $(\mathbf{x}, \mathbf{y})$ within a mini-batch, with its corresponding image representation $\mathbf{v}$ (See Section 3), the Spurious-aware Contrastive Loss is formulated as follows:

$$\mathcal{L}_{\text{SCL}}(\mathbf{x}) = -\sum_{p=1}^{P} \log \frac{\exp(\mathbf{v}^\top \mathbf{v}_p / \tau)}{\exp(\mathbf{v}^\top \mathbf{v}_p / \tau) + \sum_{j=1}^{J} \exp(\mathbf{v}^\top \mathbf{v}_j / \tau)}, \tag{3}$$

where $\{\mathbf{v}_p\}_{p=1}^{P}$ are the representations of images from the same group as $\mathbf{x}$, and $\{\mathbf{v}_j\}_{j=1}^{J}$ are the representations of the images from different groups. This loss $\mathcal{L}_{\text{SCL}}$ is essentially a supervised contrastive loss at the group level, designed to enhance the similarity of representations within the same group while distinguishing those from different groups.

**Contrastive Cosine Similarity.** Building upon our exploration of spurious-aware loss functions, we introduce the Contrastive Cosine Similarity loss, a method that aligns with the SCL objective. This loss function calculates the differences in cosine similarity between positive and negative pairs within a mini-batch. Specifically, for a given image $\mathbf{x}$, the Spurious Cosine Similarity (SCS) loss is given as:

$$\mathcal{L}_{\text{SCS}}(\mathbf{x}) = -\sum_{p=1}^{P} \frac{\mathbf{v}^\top \mathbf{v}_p}{\|\mathbf{v}\| \cdot \|\mathbf{v}_p\|} + \sum_{j=1}^{J} \frac{\mathbf{v}^\top \mathbf{v}_j}{\|\mathbf{v}\| \cdot \|\mathbf{v}_j\|}, \tag{4}$$

where $\{\mathbf{v}_p\}_{p=1}^{P}$ and $\{\mathbf{v}_j\}_{j=1}^{J}$ are the representations of images within the same group and different groups with respect to $\mathbf{x}$, respectively (identical definition as in $\mathcal{L}_{\text{SCL}}$). This loss straightforwardly encourages the minimization of the similarity between features from different groups, while maximization within the same group.

**Training Objective.** In our study, we examine both of the mentioned loss functions to encourage group-level contrastive representation learning. Additionally, we include the standard CLIP loss (Radford et al., 2021; Yang et al., 2023a), denoted by $\mathcal{L}_{\text{CLIP}}$, to stabilize the fine-tuning process of CLIP. The overall loss function is formulated as:

$$\mathcal{L}_{\text{Total}} = \mathcal{L}_{\text{Spurious}} + \mathcal{L}_{\text{CLIP}}, \tag{5}$$

where $\mathcal{L}_{\text{Spurious}}$ is selected as either $\sum_{\mathbf{x}} \text{w}(\mathbf{x}, \mathbf{y}) \mathcal{L}_{\text{SCL}}(\mathbf{x})$ or $\sum_{\mathbf{x}} \text{w}(\mathbf{x}, \mathbf{y}) \mathcal{L}_{\text{SCS}}(\mathbf{x})$. Of note, in Section 5.2, we undertake an ablation study to compare the efficacy of the two loss function choices.

## 4.3 FEATURE DROPOUT FOR MITIGATING REPRESENTATION COLLAPSE

Recent studies (Kim et al., 2023; Gal & Ghahramani, 2016; Li et al., 2023b; Shi & Yang, 2023) have demonstrated a potential issue with VLMs – the tendency for visual representations to overfit textual features during training. This issue is particularly pronounced in fine-tuning, where the neural networks often tend to give priority to learning "easy-to-learn" spurious attributes instead of concentrating on the true label information, especially when tuning only a small fraction (*e.g.*, approximately 1%) of the model's parameters.

---

[2]The raw value of w can be huge when $\delta$ is small. Thus we adopt a truncation $\text{w} \leftarrow \min\{\text{w}, W\}$ to adjust its scale in our experiments and also to satisfy the assumption in Theorem 3.

Table 1: **Comparison of results across method and benchmarks.** We evaluate various supervised, semi-supervised, and our proposed SFR, on multiple benchmark datasets: Waterbirds, CelebA, CheXpert, and MetaShift. Best and second-best results among supervised methods are shown in blue and red, respectively. For a detailed discussion, please refer to Section 5.1.

| | | ResNet-50 | | | | | | | | ViT | | | | | | | |
| | | Waterbirds | | CelebA | | CheXpert | | MetaShift | | Waterbirds | | CelebA | | CheXpert | | MetaShift | |
| | Method | WGA | Avg | WGA | Avg | WGA | Avg | WGA | Avg | WGA | Avg | WGA | Avg | WGA | Avg | WGA | Avg |
|---|---|---|---|---|---|---|---|---|---|---|---|---|---|---|---|---|---|
| supervised | ERM | 46.92 | 93.73 | 53.32 | 94.07 | 18.36 | 90.31 | 73.78 | 90.35 | 69.47 | 97.74 | 23.33 | 94.30 | 14.07 | 90.48 | 90.77 | 97.37 |
| | GroupDRO (Sagawa et al., 2020) | 74.55 | 84.79 | 84.09 | 89.54 | 65.96 | 71.77 | 80.63 | 87.99 | 89.88 | 96.92 | 86.77 | 88.01 | 67.10 | 72.33 | 92.31 | 97.03 |
| | SaC (Yang et al., 2023a) | 77.48 | 84.28 | 81.11 | 91.10 | 65.36 | 72.34 | 80.00 | 89.02 | 88.63 | 96.92 | 86.11 | 90.05 | 64.88 | 73.85 | 92.31 | 96.91 |
| | DFR (Kirichenko et al., 2023) | 73.42 | 83.52 | 81.67 | 91.61 | 60.64 | 74.96 | 78.01 | 88.44 | 88.47 | 97.60 | 86.16 | 88.77 | 62.84 | 71.12 | 91.26 | 97.03 |
| | SFR (ours) | 79.42 | 84.36 | 87.78 | 89.53 | 66.92 | 75.18 | 83.08 | 89.59 | 90.50 | 96.79 | 88.89 | 90.80 | 65.01 | 74.42 | 93.85 | 97.03 |
| semi-sup | AFR (Qiu et al., 2023) | 48.32 | 89.05 | 74.40 | 85.42 | 48.09 | 65.06 | 74.62 | 80.36 | 73.33 | 88.13 | 69.99 | 85.17 | 35.07 | 79.99 | 89.98 | 97.14 |
| | JTT (Liu et al., 2021) | 61.75 | 90.82 | 80.56 | 87.63 | 45.96 | 65.40 | 73.30 | 87.76 | 87.38 | 97.30 | 73.33 | 93.77 | 43.09 | 74.90 | 89.38 | 90.85 |
| | CnC (Zhang et al., 2022b) | 61.37 | 87.34 | 80.89 | 88.82 | 46.92 | 71.86 | 77.49 | 88.56 | 84.47 | 97.34 | 81.67 | 93.33 | 59.25 | 69.05 | 91.58 | 94.51 |

To address this challenge, we propose a simple yet effective feature dropout strategy. Specifically, as described in Section 3, $\mathbf{v}$ denotes the image representation outputted by the vision branch's projection head in CLIP. In implementation, during training, we randomly drop a predetermined proportion $p$ of $\mathbf{v}$'s entries, with this proportion being a tunable hyper-parameter. In other words, during the training phase, the deactivation of feature $\mathbf{v}$ adheres to a `Bernoulli`$(p)$ distribution. Conversely, in the inference phase, dropout is not applied, ensuring that all neurons remain active. Our proposed SFR, which synergizes feature dropout with sample reweighting, has achieved SoTA results in boosting group robustness for fine-tuning CLIP. This is substantiated by our extensive experimental findings and thorough ablation studies, as detailed in Section 5.2.

# 5 EXPERIMENTS

In this section, we present our experimental results of SFR on diverse benchmarks. The benchmark datasets and implementation details are described in Appendix A and B. Further theoretical analysis are provided in Appendix E.

## 5.1 MAIN RESULTS

**Baselines.** In our study, we begin by overviewing six representative methods that address spurious correlations. These methods encompass both supervised and semi-supervised approaches. (1) *supervised*: **GroupDRO** (Sagawa et al., 2020), a supervised method, directly utilizes Worst Group Accuracy (WGA) across predefined groups as its training goal, leveraging group annotations in the data. This choice of training objective closely aligns with the evaluation metric, contributing to its effectiveness. **DFR** (Kirichenko et al., 2023) builds on the observation that features learned with ERM capture both spurious and core features. It employs a group-balanced dataset to fine-tune the final classification layer, thereby achieving performance on par with GroupDRO (Sagawa et al., 2020). **Spurious-aware Contrastive Learning (SaC)** (Yang et al., 2023a), to the best of our knowledge, is the first work on addressing spurious correlations in multi-modal foundation models, *e.g.*, CLIP, which only fine-tunes the representation of pre-trained CLIP using spurious-aware contrastive learning. This essentially involves supervised contrastive learning with group labels, contributing to its effectiveness. (2) *semi-supervised*: **AFR** (Qiu et al., 2023) begins by training an ERM model to infer information about minority groups. It then retrains the model's last layer on a reweighted dataset, where the weight is designed to capture the confidence of the first-stage model. **JTT** (Liu et al., 2021) employs a two-stage design, where the first stage involves learning an ERM model to select a subset of misclassified data, which is then used together with the entire training data to retrain the model. **CnC** (Zhang et al., 2022b) first trains an ERM model, and then uses contrastive learning to train a target model, where the positive pairs are selected from points within the same class but different predictions by the ERM model, and the negative pairs are chosen from points within different class but same predictions.

**Analysis.** Evaluation results are summarized in Table 1 and Figure 4, where all models are compared under the same experimental setting. More specifically, we retain the vision encoder and language encoder of CLIP as frozen, updating only the projection layer of the vision branch (Yang et al., 2023a). This approach allows for efficient training and mitigation of spurious correlations. The following observations can be drawn: (1) Our SFR demonstrates superior performance compared to all other

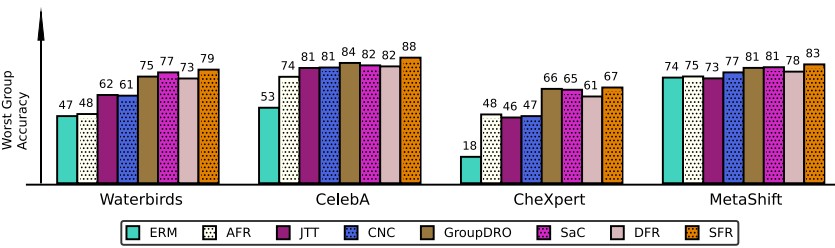

Figure 4: **Comparison of results across method and benchmarks using the CLIP-ResNet50 architecture.** We report the Worst Group Accuracy as the performance metric. The evaluated methods include ERM, semi-supervised baselines (*i.e.*, AFR (Qiu et al., 2023), JTT (Liu et al., 2021), CnC (Zhang et al., 2022b)), supervised baselines (*i.e.*, GroupDRO (Sagawa et al., 2020), SaC (Yang et al., 2023a), DFR (Kirichenko et al., 2023)), and SFR (ours). As shown, we observe that SFR outperforms all baselines across all benchmarks (*i.e.*, Waterbirds, CelebA, CheXpert, and MetaShift).

supervised or semi-supervised methods across nearly all benchmarks. SFR with ResNet-50 obtains $\{1.94 \sim 6.00, 3.96 \sim 6.67, 0.96 \sim 6.28 \ 2.45 \sim 5.07\}$ WGA performance boosts over GroupDRO, SaC, DFR, on Waterbirds, CelebA, CheXpert, and MetaShift, respectively. This validates the effective of SFR. When using ViT backbone, SFR has achieved competitive performance on Waterbirds, CelebA, and MetaShift. (2) As is shown in Table 1, we observe that the improvements are more pronounced using ResNet backbone than ViT backbone. The possible reason is that, with the aid of multi-modality information during fine-tuning, using ViT is more robust towards spurious correlation features, as echoed in recent work (Ghosal & Li, 2023). (3) The training-validation curve[3] (Figure 5) compares SFR with various supervised baselines. This visualization shows that SFR not only achieves a more optimal solution but also does so with a faster convergence rate. Clearly, SFR demonstrates consistent advantages over all other baselines in the Waterbirds, CelebA, and MetaShift datasets. On CheXpert, SFR performs on par with the SoTA method, GroupDRO. This observation underscores the efficiency of our method in achieving superior performance. (4) From Figure 7 (in Appendix), we provide the training-validation curve of SFR for all four groups within the Waterbirds dataset. It is observed that SFR quickly adapts to the two majority groups (*i.e.*, "landbird on land" and "waterbird on water"). Although its convergence is somewhat slower on the two minority groups, it ultimately yields high accuracy in these groups, demonstrating its effectiveness in handling diverse group categories.

## 5.2 ABLATION STUDIES

We perform a comprehensive analysis of each key component in our SFR. Additional ablations are provided in Appendix D. All ablations are conducted on Waterbirds, unless explicitly stated otherwise. Moreover, all other hyperparameters remain the same across these experiments.

**Ablation on w/ vs. w/o Feature Dropout.** We first analyze the importance of the feature dropout module using CLIP-ResNet with and without dropout. Results are collected in Table 3 (in Appendix). As shown, we observe that feature dropout brings a significant 1.59 WGA improvement compared to the scenarios without feature dropout. This confirms the effectiveness of feature dropout in improving group robustness. We speculate that our feature dropout can induce the implicit regularization to mitigate spurious correlations.

**Ablation on Diverse Dropout Strategies.** An appropriate design of dropout methods determines the achievable performance of our SFR. In our study, we select two representatives among various dropout variants, Concrete Dropout (Gal et al., 2017) and DropBlock (Ghiasi et al., 2018). Results are summarized in Table 3 (in Appendix). We observe that standard dropout achieves 3.77 and 9.04 WGA gains compared to the other two variants.

**Ablation on Different Locations for Dropout.** To demystify the optimal placement of dropout, we investigate an alternative dropout location and record the score in Table 3 (in Appendix). We primarily consider two options: (1) Inserting dropout *before* the projection head, and (2) Implementing it *after* the projection head (*i.e.* the current configuration). The result indicates that applying dropout after

---

[3]We evaluate the performance using WGA on the validation dataset during the training process.

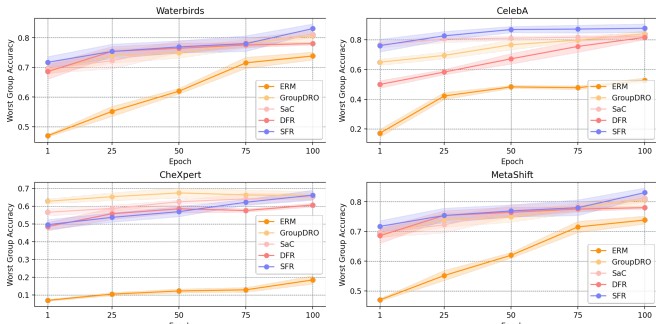

Figure 5: **Training-validation curves across various methods using CLIP-ResNet50 backbone.** We utilize WGA to evaluate the performance on a hold-out validation dataset at 25-epoch intervals during the entire training period. Our proposed SFR utilizes the $\mathcal{L}_{\text{SCS}}$ loss, as detailed in Sec. 4.2. Results are averaged over three different random seeds.

the projection head yields much better results than the alternative placement before it. A plausible explanation is that the preservation of common features from the frozen visual encoders are essential for training. This finding further supports that CLIP can capture the core features applicable to diverse downstream tasks through pre-training. It underscores the efficacy of adopting a fine-tuning strategy in effectively mitigating spurious correlations, further validating this approach in practicality.

**Extra Study.** More investigations about (1) different dropout ratio in Appendix D.1; (2) different loss functions in Appendix D.2; (3) different architectures are in Appendix D.3; and (4) theoretical analysis in Appendix E.

## 6 CONCLUSION

We introduced SFR, a simple yet effective robust multi-modal learning framework that greatly improve group robustness across multiple benchmarks. We develop disagreement-based importance weights, assigning unique weights to each instance in the training data, effectively addressing the variance in instance-level learning challenges. We further verify representation collapse in multi-modal model training. To counter this, we incorporate dropout to better regulate training for majority groups while promoting focus on minority groups. We believe our approach holds wider implications for group robustness, offering a new perspective on applying regularization to reduce spurious correlations in multi-modal foundation models. This method has the potential for broader applications, including addressing various spurious attributes and being integrated into language foundation model training.

**Limitations.** Defending machine learning models from spurious correlations holds immense promise for developing more reliable and trustworthy medical AI. Our proposed SFR framework significantly enhances group robustness in multi-modal learning, paving the way for practical implementations across a broad spectrum of real-world biomedical applications. Additionally, as part of our future research direction, it's crucial to tackle challenges related to fairness and privacy within the domain of spurious correlations, ensuring that these advanced AI solutions are both equitable and secure.

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
