# A    ADDITIONAL EXPERIMENTAL DETAILS

**Datasets.**    We mainly use four image classification benchmarks:

- **Waterbirds** (Wah et al., 2011): A popular binary classification dataset to address spurious correlations. This dataset integrates Caltech-UCSD Birds-200-2011 (CUB) (Wah et al., 2011) with backgrounds derived from Places dataset (Zhou et al., 2017). The objective is to classify images as landbirds or waterbirds, potentially influenced by spurious background attributes of land or water. Our data partitioning is consistent with (Idrissi et al., 2022).

- **CelebA** (Liu et al., 2015): A comprehensive dataset encompassing over 200,000 celebrity portraits. A central task, pivotal in exploring spurious correlations, aims to identify hair color, particularly differentiating blond from non-blond. Notably, gender becomes an unintended influential attribute. Our dataset splits align with (Idrissi et al., 2022). The dataset adheres to the *Creative Commons Attribution 4.0 International* license.

- **CheXpert** (Irvin et al., 2019): A collection of over 200,000 chest X-ray images sourced from Stanford University Medical Center. The primary classification, "No Finding", denotes a healthy assessment. Taking cues from (Seyyed-Kalantari et al., 2021), we incorporate both race and gender as influential variables. We follow the partitioning set by (Yang et al., 2023b).

- **MetaShift** (Liang & Zou, 2022) is a versatile approach to generating high-quality image datasets, making use of the Visual Genome project (Krishna et al., 2017). In our study, we follow (Yang et al., 2023b) to utilize the pre-processed Cat *vs.* Dog dataset, with the goal of distinguishing between the two animal. It is crucial to acknowledge the dataset's inherent challenge: a spurious attribute associated with the image background, often placing cats indoors and dogs outdoors. We have selected the "unmixed" version of the dataset, directly sourced from the original authors' codebase.

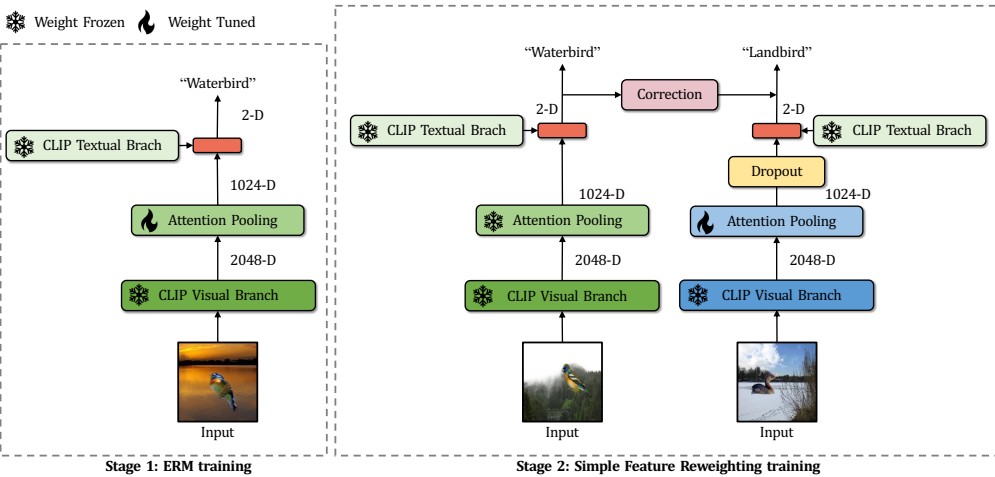

Figure 6: **The overview of the proposed SFR framework.** In Stage 1, we utilize standard Empirical Risk Minimization (ERM) with a cross-entropy loss to train a CLIP model. Following this, Stage 2 involves fine-tuning a new CLIP model of the same architecture, but with an emphasis on learning representations invariant to spurious attributes. This is achieved using disagreement-based importance weighting (See Sec. 4.1), which are calculated based on entropy disagreement between the ERM and our target model, and dropout (See Sec. 4.3). Additionally, we focus solely on fine-tuning the CLIP vision branch's last-layer projection head using a contrastive loss (See Sec. 4.2) on the classifier output and the truth class labels. Dimensions specified pertain to CLIP using the ResNet-50 backbone and Waterbirds.

Table 2: **Overview of Experimental Settings.** We provide a comprehensive overview of the experimental settings, including the model architecture, training processes, evaluated methods, and data preprocessing.

| Condition | Parameter | Value |
|---|---|---|
| *Model Architecture*: | | |
| CLIP-RN50 (Radford et al., 2021) | Input size | 256×256 |
| CLIP-ViT (Radford et al., 2021) | Input size | 336×336 |
| *Training*: | | |
| Optimizer | Type | SGD |
| | Learning rate | 1e-5 |
| | Momentum | 0.9 |
| | L2 weight decay | 1e-4 |
| | Metric to pick best model | WGA |
| *Algorithm-specific*: | | |
| ○ SFR (ours) | Weight computing start epoch | 10 |
| | Weight computing start epoch | 5 |
| | Dropout on feature | 0.1 |
| | Coefficient of weight | 0.25 |
| CnC (Zhang et al., 2022b) | Number of positive points | 16 |
| | Number of negative points | 16 |
| JTT (Liu et al., 2021) | $\lambda_{up}$ | 10 |
| GroupDRO (Sagawa et al., 2020) | $\eta$ | 0.01 |
| *Dataset-specific*: | | |
| Waterbirds (Wah et al., 2011) | Raw input size | $224 \times 224$ |
| | Reweight range | $(0.8, 2.0)$ |
| CelebA (Liu et al., 2015) | Raw input size | $178 \times 218$ |
| | Reweight range | $(0.5, 5.0)$ |
| CheXpert (Irvin et al., 2019) | Raw input size | $390 \times 320$ |
| | Reweight range | $(0.2, 2.0)$ |
| MetaShift (Liang & Zou, 2022) | Raw input size | $256 \times 256$ |
| | Reweight range | $(0.8, 2.0)$ |

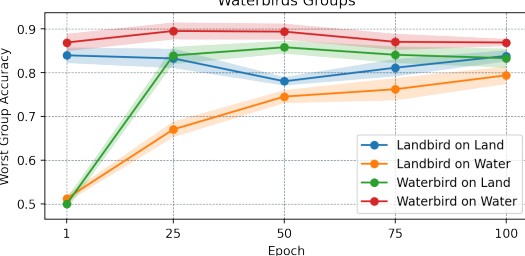

Figure 7: **Training-validation curves using SFR with CLIP-ResNet50 backbone.** The result is reported for each of the four groups of Waterbirds dataset. Accuracy is evaluated on a hold-out validation dataset at 25-epoch intervals during the entire training process. SFR utilizes the $\mathcal{L}_{SCS}$ loss, as detailed in Sec. 4.2. Results are averaged over three different random seeds.

## B  IMPLEMENTATION DETAILS

In this section, we provide the full implementation details for all evaluated methods. We use the hyperparameter details that are given in Table 2.

**Model Architectures.**  Within CLIP, we adopt two major visual backbones: ResNets (RN) and Visual Transformers (ViT). Their performance intricacies are thoroughly explored in (Radford et al., 2021). In our study, ResNet-50 (RN50) and ViT-L/14@336px, tailored for 336x336 pixel images, stand as the primary representatives, aligning with the setting in (Yang et al., 2023a). Meanwhile, for the language branch, we adopt the pre-trained mask language model, BERT (Kenton & Toutanova, 2019), due to its SoTA performance.

Our methods, together with the baseline methods, consistently retain the language and visual encoders, with an emphasis on fine-tuning the projection layers. This closely aligns with the established protocols from prior work (Yang et al., 2023a) and ensures a fair comparison. Furthermore, it is deliberately aimed at developing methods that are efficacious as well as computationally efficient.

**Metrics.** Worst-Group Accuracy (WGA) stands as a widely adopted metric in the spurious correlation literature (Liu et al., 2021; Kirichenko et al., 2023; Zhang et al., 2022b; Zhang & Ré, 2022; Yang et al., 2023a; Qiu et al., 2023) due to its straightforward connection with the model's robustness across different groups in the data. Specifically, WGA denotes the lowest classification accuracy observed across all groups within the test dataset, with groups defined as the Cartesian product of all target classes and all spurious attributes. A high WGA indicates that the model performs well even on minority groups, showcasing robustness against spurious correlations across different subsets. Therefore, we adopt WGA as our primary performance indicator.

In parallel, Average Accuracy represents the classification accuracy averaged over all groups within the test set. To be specific, the Average Accuracy reported in this paper, *i.e.*, 'Avg' in Table 1 (Main Context) and Appendix Table 5, is suggested by (Sagawa et al., 2020), which is a weighted average of the test accuracy of the groups. The weight for each group is proportional to their sizes in the training set. This metric provides an integrated perspective on the model's overall performance across all class categories. It is worth noting from Table 1 (Main Context) that despite the significant WGA improvement over ERM, the Average Accuracy drops by a little bit. We point out that this issue of mediocre Average Accuracy is a common limitation in most existing methods across supervised and semi-supervised settings, which is evident in others (Sagawa et al., 2020; Zhang & Ré, 2022; Yang et al., 2023a; Kirichenko et al., 2023). As seen from these influential works, no method uniformly outperforms others in Average Accuracy across every benchmark dataset.

**Experimental Setup.** In each of our experimental trials, we maintained a uniform experimental configuration, employing a single NVIDIA GeForce RTX 3090 GPU and the fixed random seeds for consistency. The experiments were conducted utilizing PyTorch version 1.10.2+cu113 and Python 3.8.11, to guarantee reproducibility across our investigations.

**Reweight Computation.** Here, we elucidate the methodology employed for generating the weights associated with each dataset—a pivotal aspect of our research. The computation of reweights necessitates the utilization of an ERM-tuned CLIP model. For every data point within the training set, we leverage the ERM-tuned CLIP model to derive an initial prediction. Throughout the training epochs, the current model weights are adopted to obtain a contemporaneous prediction. The specific starting epoch and update frequency are outlined in Table 2. Employing these two predictions, we input them into Eqn. 1 (Main Context) and Eqn. 2 (Main Context) to calculate an initial weight for each data point. To prevent excessively high weight values and ensure their validity, we apply a coefficient, upper-bound, and lower-bound to confine the computed weights within a specified range. Comprehensive information regarding the hyperparameters is available in Table 2.

**Training Details.** For all methods assessed in our experiments, encompassing both baseline models and our proposed approach, we employ an SGD optimizer with a weight decay set to $10^{-4}$ and a momentum set to $0.9$. The learning rate is held constant at $10^{-5}$ throughout training, which spans 100 epochs. The model selection process remains consistent across all methods. At the end of each epoch, we assess the model's performance on the validation set, opting for the one that exhibits the best worst-group accuracy for final testing. All accuracy metrics presented in this paper are derived from evaluations on the test set.

**Dataset Preprocessing.** The dataset preprocessing steps remain consistent across all four datasets and the various methods under evaluation. Initially, we resize the raw images while preserving a fixed height-to-width ratio. This ensures that the shorter edge of the image attains dimensions of 256 for ResNet-50 and 336 for ViT-L/14@336px. Subsequently, the resized image undergoes cropping to dimensions of 256×256 for ResNet-50 and 336×336 for ViT-L/14@336px. Following this resizing and cropping, the image is normalized through the subtraction of the average pixel value and division by the standard deviation, a process consistent with CLIP (Radford et al., 2021). No additional data augmentation is applied beyond these steps, as our methods primarily emphasize lightweight fine-tuning, involving updates to the projection layer of the vision branch in CLIP. Introducing data augmentation in this context could potentially result in underfitting due to the modest parameter size of the projection layer.

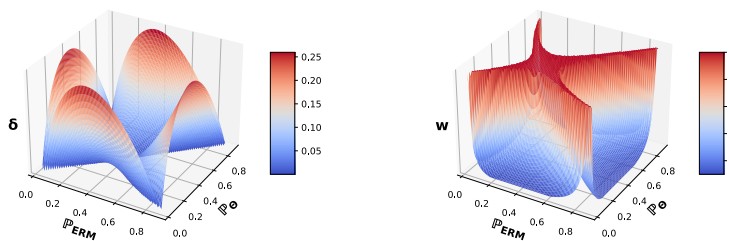

Figure 8: **Values of disagreement score $\delta$ (left) and sample weight w (right) as a function of $\mathbb{P}_{\mathbf{ERM}}$ and $\mathbb{P}_{\theta}$.** Since the raw weight $w = 1/\delta$ can be very large when $\delta$ is small, we truncate it by $w \leftarrow \min\{w, 5\}$ in the plot.

Table 3: **Ablation on dropout variants and appropriate locations to insert dropout on Waterbirds.** We adopt the CLIP-ResNet50 as backbone with $\mathcal{L}_{\text{SCS}}$. For our SFR, we apply dropout strategy to features after the final projection head of CLIP's visual branch. Please refer to the main text for detailed discussions.

|  | Options | WGA | Avg |
|---|---|---|---|
|  | ○ SFR (ours) | 79.42 | 84.36 |
| Dropout variants | Concrete Dropout (Gal et al. 2017) | 75.65 | 85.93 |
|  | DropBlock (Ghiasi et al. 2018) | 70.38 | 84.36 |
|  | w/o Dropout | 77.83 | 83.67 |
| Dropout location | Before Projection Head | 62.13 | 82.31 |

## C ADDITIONAL RESULTS

**Training curve.** From Figure 7 (in Appendix), we provide the training-validation curve of SFR for all four groups within the Waterbirds dataset. It is observed that SFR quickly adapts to the two majority groups (*i.e.*, "landbird on land" and "waterbird on water"). Although its convergence is somewhat slower on the two minority groups, it ultimately yields high accuracy in these groups, demonstrating its effectiveness in handling diverse group categories.

**Disagreement score and sample weight.** In Figure 8 (in Appendix), we depict the values of the disagreement score $\delta$ and the corresponding sample weight w as functions of $\mathbb{P}_{\text{ERM}}$ and $\mathbb{P}_{\theta}$, as introduced in Section 4.1, Eqn. 1 and 2. It is noteworthy that the score $\delta$ tends to be large when two models exhibit significant disagreement in their certainty. For instance, when $\mathbb{P}_{\text{ERM}}$ approaches 0 or 1 while $\mathbb{P}_{\theta}$ is close to 0.5, $\delta$ reaches its maximum, and vice versa. Consequently, the weight w attains higher values when both models possess a similar level of certainty at the sample, even if their certainties are in opposite directions.

Additionally, in the right plot of Figure 8 (in Appendix), we display the truncated weight $w \leftarrow \min\{w, 5\}$ since the raw weight $w = 1/\delta$ can be substantial for small $\delta$. We apply this type of truncation in all our experiments on SFR, with $w \leftarrow \min\{w, W\}$, where $W > 0$ is a dataset-specific hyperparameter as introduced in Table 2.

## D ABLATION STUDY

### D.1 ABLATION ON DROPOUT RATIO.

We test the choice of the dropout ratio in Table 4. We find that $p = 0.1$ is optimal.

### D.2 ABLATION ON DIFFERENT LOSS FUNCTIONS.

We conduct a further study of loss functions (*i.e.*, $\mathcal{L}_{\text{SCL}}$ and $\mathcal{L}_{\text{SCS}}$) by examining CLIP using ResNet50 and ViT as backbone on all four benchmarks (*i.e.*, Waterbirds, CelebA, CheXpert, and MetaShift).

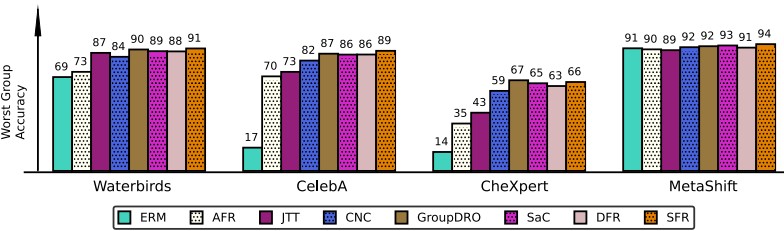

Figure 9: **Comparison of results across method and benchmarks using the CLIP-ViT architecture.** We report the Worst Group Accuracy as the performance metric. As shown, we observe that SFR outperforms all baselines across almost all benchmarks (*i.e.*, Waterbirds, CelebA, and MetaShift). On CheXpert, GroupDRO slightly outperforms SFR.

Table 4: **Ablation on dropout ratio** $p$**.** We adopt the CLIP-ResNet50 as backbone with $\mathcal{L}_{\text{SCS}}$.

| Dropout ratio $p$ | WGA | Avg |
|---|---|---|
| 0.05 | 78.23 | 84.74 |
| ◦ 0.1 (ours) | 79.42 | 84.36 |
| 0.2 | 77.69 | 84.22 |
| 0.5 | 75.65 | 82.27 |

The comparison results of {SaC with $\mathcal{L}_{\text{SCL}}$ and $\mathcal{L}_{\text{SCS}}$} and our {SFR with $\mathcal{L}_{\text{SCL}}$ and $\mathcal{L}_{\text{SCS}}$} are reported in Table 5 (in Appendix). We observe that using $\mathcal{L}_{\text{SCS}}$ consistently performs better than $\mathcal{L}_{\text{SCL}}$. Specifically, using CLIP-ResNet50, we see that SFR with $\mathcal{L}_{\text{SCS}}$ obtains {1.38, 0.71, 1.54} gain in WGA compared to $\mathcal{L}_{\text{SCL}}$ across Waterbirds, CelebA and MetaShift, respectively, but slightly underperforms on CheXpert. Meanwhile, we find that SFR consistently outperforms SaC across all four benchmarks, suggesting an enhanced group robustness.

Table 5: **Ablation on different spurious loss terms.** We evaluate two spurious-aware loss terms, $\mathcal{L}_{\text{SCS}}$ and $\mathcal{L}_{\text{SCL}}$, on SFR (ours) and SaC (Yang et al., 2023a). Please see Sec. 4.2 for detailed discussion of the loss terms, and refer to the main text for the discussion of the results.

| | ResNet-50 | | | | | | | | ViT | | | | | | | |
|---|---|---|---|---|---|---|---|---|---|---|---|---|---|---|---|---|
| | Waterbirds | | CelebA | | CheXpert | | MetaShift | | Waterbirds | | CelebA | | CheXpert | | MetaShift | |
| Method | WGA | Avg | WGA | Avg | WGA | Avg | WGA | Avg | WGA | Avg | WGA | Avg | WGA | Avg | WGA | Avg |
| SaC(with $\mathcal{L}_{\text{SCS}}$) (Yang et al., 2023a) | 77.48 | 84.28 | 81.11 | 91.10 | 65.36 | 72.34 | 80.00 | 89.02 | 88.63 | 96.92 | 86.11 | 90.05 | 64.88 | 73.85 | 92.31 | 96.91 |
| SaC(with $\mathcal{L}_{\text{SCL}}$) (Yang et al., 2023a) | 75.21 | 86.62 | 82.38 | 90.65 | 62.77 | 74.11 | 81.09 | 87.87 | 88.94 | 96.02 | 85.15 | 89.52 | 63.22 | 73.02 | 92.85 | 97.03 |
| ◦ SFR (with $\mathcal{L}_{\text{SCS}}$) | 79.42 | 84.36 | 87.78 | 89.53 | 66.21 | 70.44 | 83.08 | 89.59 | 90.50 | 96.59 | 88.89 | 90.80 | 65.01 | 74.42 | 93.85 | 97.03 |
| ◦ SFR (with $\mathcal{L}_{\text{SCL}}$) | 78.04 | 85.33 | 87.07 | 89.20 | 66.92 | 75.18 | 81.54 | 88.56 | 90.65 | 96.74 | 87.78 | 90.99 | 65.81 | 73.83 | 92.31 | 96.97 |

### D.3 ADDITIONAL ABLATIONS ON DIFFERENT ARCHITECTURES.

In Figure 9 (in Appendix), we compare the performance of all evaluated methods on all benchmarks, using the CLIP-ViT architecture. The results of using CLIP-ViT are provided in Table 1 (Main Context). All experiment details are the same as in Appendix Table 2 (in Appendix). On most benchmarks, our SFR performs better than the other supervised and semi-supervised models when trained with CLIP-ViT. These results further demonstrate the robustness of SFR against spurious correlations.

## E ADDITIONAL THEORETICAL ANALYSIS

In this section, we present a comprehensive theoretical analysis of our disagreement-based reweighting score. Let $\mathcal{L} : \mathcal{Y} \times \mathcal{Y} \to [0,1]$ be a loss function and $f : \mathcal{X} \to \mathcal{Y}$ be the target labeling function. We also denote by $\mathcal{H}$ the hypothesis set used by the learning algorithm and by $Pdim(U)$ the pseudo-dimension of a real-valued function class $U$ (Pollard, 2012). For any hypothesis $h \in \mathcal{H}$, the generalization error on target distribution, $R(h)$, and the weighted empirical loss on source distribution, $\hat{R}_{\text{w}}(h)$, are defined as follows:

$$R(h) = \mathbb{E}_{x \sim \mathcal{P}}[\mathcal{L}(h(x), f(x))] \qquad \hat{R}_{\text{w}}(h) = \frac{1}{m} \sum_{i=1}^{m} w(x_i)\mathcal{L}(h(x_i), f(x_i)), x_i \sim \mathcal{Q}. \qquad (6)$$

where $m$ stands for sampling size, and the samples weights $\mathrm{w}(x)$ along with Rényi divergences $d_2(\mathcal{P}||\mathcal{Q})$ are defined as follows(Cortes et al., 2010):

$$\mathrm{w}(x) = \frac{\mathcal{P}(x)}{\mathcal{Q}(x)} \qquad d_2(\mathcal{P}||\mathcal{Q}) = \sum_{x \in X} \mathcal{P}(x)\left(\frac{\mathcal{P}(x)}{\mathcal{Q}(x)}\right) = \sum_{x \in X} \mathrm{w}(x)\mathcal{P}(x). \tag{7}$$

As established by Theorem 3 in (Cortes et al., 2010), the upper bound of $R(h)$, and $\hat{R}_{\mathrm{w}}(h)$, is expressed as follows. This bound provides a theoretical guarantee for our model, elucidating the relationship between hypothesis performance on source and target distributions, and underscoring the impact of distribution shifts and the efficacy of the reweighting strategy.

**Theorem 3 ((Cortes et al., 2010))** *Let $\mathcal{H}$ be a hypothesis set such that $Pdim(\mathcal{L}_h(x) : h \in \mathcal{H} = p < +\infty)$. Assume that $d_2(\mathcal{P}||\mathcal{Q}) < +\infty$ and $\mathrm{w}(x) \neq 0$ for all x. Then for $0 < \delta < 1$. With probability at least $1 - \delta$ ,it holds that:*

$$\left|R(h) - \hat{R}_{\mathrm{w}}(h)\right| \leq 2^{5/4}\sqrt{d_2(\mathcal{P}||\mathcal{Q})}\left(\frac{p\log\frac{2me}{p} + \log\frac{4}{\delta}}{m}\right)^{3/8}. \tag{8}$$

### E.1 A DEEP LOOK INTO THEOREM 3

According to the right side of Theorem 3, the upper bound is influenced by the sampling size $m$ and the Rényi divergence $d_2(\mathcal{P}||\mathcal{Q})$. For a fixed $m$, the Rényi divergence is pivotal in determining the bound, significantly affecting the performance of importance weighting. This highlights the importance of properly estimating distributional differences to ensure robust model performance; otherwise, the model may suffer from increased generalization error and reduced effectiveness in real-world applications. Particularly, in the deep learning era, where high-dimensional and complex data distributions prevail, explicitly measuring the distance between two distributions often leads to infinite $\mathrm{w}(x)$ (Byrd & Lipton, 2019), thereby violating the assumptions of Theorem 3.

### E.2 ANALYSIS OF DISAGREEMENT-BASED REWEIGHTING SCORE

Leveraging importance sampling in practice, on the one hand, offers straightforward yet effective benefits; on the other hand, several studies have identified potential challenges,(e.g. unbounded reweighting score $\mathrm{w}(x)$), particularly within the framework of modern deep learning. Robert et al.(Robert, 1999) discussed a phenomenon called weight degeneracy, which occurs during the iterative approximation of the target distribution, as certain weights significantly diminish due to the inadequate selection of bridging density functions. Bugallo et al.(Bugallo et al., 2017) highlighted that existing methods employing a Bayesian approach to model target distributions are hindered by significant computational overhead and increased time complexity. Byrd et al.(Byrd & Lipton, 2019) indicates that importance weighting becomes ineffective with complex data structures due to high-variance estimates in high-dimensional contexts.

Our proposed method tackles these challenges in different ways. *i*) Eqn. 1 shows that our designed disagreement-based reweighting score is data-agnostic, requiring no task-specific experience. This allows for easy implementation across different scenarios without concerns of weight degeneracy. *ii*) Our proposed approach eliminates the need for an iterative process to measure the target distribution, resulting in a time-efficient approach, which is especially beneficial for handling large-scale datasets. *iii*) Our method circumvents the high-variance issues typically encountered with high-dimensional data by implicitly measuring the divergence between two distributions through an entropy function. By focusing on modeling entropy rather than the distribution itself, we achieve a more stable and robust approach, effectively managing complexity without directly modeling the intricate details of the data. Extensive experimental results in Table 1 demonstrate that our method is both effective and suitable within the deep learning framework. To sum up, our design of disagreement-based reweighting score is an effective mimic of the true $\mathrm{w}(x) = \mathcal{P}(x)/\mathcal{Q}(x)$, making it more suitable for lightweight, efficient, plug-and-play applications in modern deep learning era.