# OpenReview forum: "Robust Multi-modal Learning with Shifted Feature Reweighting Against Spurious Correlations"
_ICLR.cc/2025/Conference — ICLR 2025 Conference Withdrawn Submission_

### Official Review · Reviewer_oQky · 2024-10-23

**Soundness:** 3
**Presentation:** 1
**Contribution:** 2
**Rating:** 5
**Confidence:** 2

**Summary:**

This paper introduces a novel method called SFR (Shifted Feature Reweighting) to mitigate the reliance on spurious features in machine learning models. The core idea behind SFR is to allocate weights to the instance features, allowing the model to focus on more relevant attributes. To address potential collapse issues that may arise during the fine-tuning process, the authors employ a feature dropout technique. This dual approach not only enhances model robustness but also promotes better generalization across diverse datasets.

**Strengths:**

The paper excels in its visualization, providing clear and informative graphics that effectively illustrate the methodology and results.

Additionally, the focus on individual sample difficulty, as opposed to broad group characteristics, introduces a refreshing novelty to the existing literature. This emphasis on individual samples allows for more tailored model training, potentially leading to improved performance.

Furthermore, the approach is supported by both empirical validation and theoretical analysis, which enriches our understanding of the underlying mechanisms. This theoretical grounding helps clarify the conditions under which the method is expected to excel, providing valuable insights for future research.

**Weaknesses:**

Despite its strengths, the paper lacks a comprehensive pipeline illustration, which would greatly aid in understanding the overall methodology.

Moreover, the research appears to focus exclusively on the CLIP model, raising questions about the applicability of the proposed method to other architectures. This narrow scope could limit the broader impact of the findings.

While it is mentioned that the approach is lightweight, a concrete comparison in terms of training time and resources (Flops) would be beneficial.

**Questions:**

1. There are some inaccuracies in the references, such as referencing Eq. 2 in Figure 3.

2. The color scheme used in Figure 4 is not well-coordinated, which could hinder clarity and interpretation.

3. Providing an empty GitHub repository link is a poor choice, as it fails to offer readers any supplementary resources or code to explore.

4. The absence of a clear pipeline illustration makes it difficult to follow the proposed method's implementation.

5. The study’s narrow focus on the CLIP model raises the question of its effectiveness across other models—what about the implications for different architectures?

---

### Official Review · Reviewer_K1AR · 2024-11-03

**Soundness:** 3
**Presentation:** 3
**Contribution:** 3
**Rating:** 6
**Confidence:** 4

**Summary:**

The paper introduces Shifted Feature Reweighting (SFR), a method designed to enhance the robustness of VLMs against spurious correlations, aiming to improve group robustness with minimal computational overhead. SFR employs a disagreement-based importance weighting strategy and feature dropout to prevent representation collapse during fine-tuning. This approach is evaluated across multiple benchmarks, including Waterbirds and CelebA, and is supported by some theoretical analysis.

**Strengths:**

1. Addresses an important and under-explored problem in the robustness of multi-modal models.
2. Introduces a novel disagreement-based reweighting strategy that is both practical and effective.
3. Empirical results on multiple benchmarks demonstrate improvements in robustness against spurious correlations.
4. Theoretical analysis is provided in the appendix to support the claims, adding credibility to the proposed method.

**Weaknesses:**

1. To some extent, reweighting samples for fine-tuning and using dropout for regularization seem to be fairly common approaches. In this regard, the novelty of this paper is somewhat limited.
2. The paper could benefit from a clearer explanation of the proposed method, especially for the sample reweighting. It seems to be missing a detailed and in-depth discussion about why instance-level weighting is needed and matters.
3. There is a lack of ablation studies to isolate the effects of individual components in the proposed method, e.g., disagreement-based reweighting.
4. Some parts of the text lack rigour and consistency, such as the limitations and the ablation experiments about different architectures.

**Questions:**

1. Could the authors provide a deeper explanation and experimental validation for why instance-level reweighting is more effective, perhaps by including additional ablation studies?
2. Although the paper aims to advance the robustness of multimodal models, it seems that the text modality is not addressed with robustness enhancement, but only the visual modality. I wonder if this approach could also be applied to unimodal visual pre-training models. Anyway, what specific advantages does introducing a multimodal setting offer, and why is it necessary to start from a multimodal perspective?
3. The final section on limitations does not adequately discuss the constraints but rather resembles a summary.
4. Additionally, I don’t quite understand which different architectures are involved in the ablation implementation mentioned in the main text and appendix.

---

### Official Review · Reviewer_GdML · 2024-11-03

**Soundness:** 2
**Presentation:** 2
**Contribution:** 2
**Rating:** 5
**Confidence:** 4

**Summary:**

This paper proposes SFR to improve robustness in pre-trained vision-language models by addressing spurious correlations. SFR assigns different weights to training samples based on disagreement between a baseline model and the target VLM, aiming to focus on more challenging samples and mitigate biases. Additionally, the authors introduce a feature dropout technique to prevent representational collapse during fine-tuning. Their method demonstrates improved group robustness on benchmarks like Waterbirds and CelebA.

**Strengths:**

The proposed reweighting strategy is reasonable, and the implementation looks correct to me.

**Weaknesses:**

1. Outdated settings. It is unclear which variant of CLIP model is used. I have personally tested the publicly available OpenCLIP, and find it has extremely high performance zero-shot on many common benchmarks (especially with a large model size). This raises the question whether it is still reasonable to evaluate on simple datasets such as waterbirds when using VLMs pre-trained in large-scale. I personally believe it is still relevant to study distribution shift, but it is necessary to use a more challenging setting that reveals the SOTA pre-trained VLMs weaknesses.
2. The baselines are limited. For example, I believe it's also necessary to compare with zero-shot CLIP, prompt-tuning and CLIP-adapters as baselines.

**Questions:**

How about applying previous methods that deal with distribution shift on CLIP? Will they perform well?

---

### Official Review · Reviewer_ptpz · 2024-11-04

**Soundness:** 2
**Presentation:** 3
**Contribution:** 1
**Rating:** 3
**Confidence:** 4

**Summary:**

The paper primarily studies the problem of spurious correlation in Vision-Language models such as CLIP. The authors point out that one of the initial works on mitigating spurious correlation in CLIP [1] requires fine-tuning the whole model using spurious contrastive loss which can be computationally expensive. To retain empirical performance on spurious datasets without this intensive training, the authors propose fine-tuning only the final layer. During fine-tuning, they introduce a disagreement-based reweighting strategy to quantify the difficulty of individual samples. Overall, the proposed method achieves higher worst-group performance than previous approaches.


[1] Yang, Y., Nushi, B., Palangi, H. and Mirzasoleiman, B., 2023, July. Mitigating spurious correlations in multi-modal models during fine-tuning. In International Conference on Machine Learning (pp. 39365-39379). PMLR.

**Strengths:**

1. The overall paper presentation is good, and the research problem of spurious correlations in CLIP has been well-articulated to the reader.
2. The evaluation includes both ResNet-50 and ViT architectures and covers four standard spurious-correlation benchmarks. The proposed framework leads to improvement in worst-group performance over baselines for most of the datasets.

**Weaknesses:**

1. The technical contribution of this study is not clear. The proposed approach is based on two prior works: DFR[1] which proposed the idea of fine-tuning the last linear layer and SaC[2] which introduced the spurious-aware contrastive loss. Further, in line 366, the authors stated that L_spurious can be selected as $\mathcal{L}_{\text{SCL}}$, which further confuses the reader about the contribution of this study. I request the authors to clarify the technical contributions clearly and how the proposed approach is distinct from DFR[1] and SaC[2]. Based on the current draft, my understanding is that the primary contribution is improving SaC [2] in terms of computational efficiency by limiting fine-tuning to the last layer rather than the entire network. Hence, I believe the paper’s contributions may fall short of ICLR's acceptance bar.

2. The main argument of the paper is improving efficiency compared to SaC[2] (lines 83 and 205). Hence, in Table 1, the authors should compare with the naive baseline which is zero-shot classification, and also other inference-time[3] and prompt-tuning approaches [4].

3. [Minor] It is always advisable to report mean and std, as in some cases, the improvement is within 1-2%.


To summarize the weaknesses, my major concern is regarding the technical novelty and lack of comparison with more efficient inference-time and prompt-tuning approaches. I am willing to increase my score if the authors can clarify the above concerns.


[1] Kirichenko, P., Izmailov, P. and Wilson, A.G., 2022. Last layer re-training is sufficient for robustness to spurious correlations. arXiv preprint arXiv:2204.02937.

[2] Yang, Y., Nushi, B., Palangi, H. and Mirzasoleiman, B., 2023, July. Mitigating spurious correlations in multi-modal models during fine-tuning. In International Conference on Machine Learning (pp. 39365-39379). PMLR.

[3] Adila, D., Shin, C., Cai, L. and Sala, F., 2024. Zero-Shot Robustification of Zero-Shot Models. In The Twelfth International Conference on Learning Representations.

[4] Zhang, J., Ma, X., Guo, S., Li, P., Xu, W., Tang, X. and Hong, Z., Amend to Alignment: Decoupled Prompt Tuning for Mitigating Spurious Correlation in Vision-Language Models. In Forty-first International Conference on Machine Learning.

**Questions:**

1. In Equation 2, what is the primary motivation for setting the importance weight as the inverse of the disagreement score? With the current formulation, as the disagreement with ERM decreases, the importance weight for that sample increases. But I think that would primarily incentivize the target model to be close to the ERM distribution.

---

### Official Review · Reviewer_3VkF · 2024-11-04

**Soundness:** 1
**Presentation:** 1
**Contribution:** 1
**Rating:** 1
**Confidence:** 5

**Summary:**

This paper introduces a new approach for fine-tuning pre-trained vision-language models to ensure group robustness when applied to downstream tasks.

**Strengths:**

It is easy to read.

**Weaknesses:**

I have noticed a significant degree of similarity between the figures, tables, equations, and even the overall structure of this paper and those in the CFR paper [1]. Specifically, it appears that the authors have not only replicated the structure and flow of the CFR paper without citation but also omitted any acknowledgment of CFR's results, substituting them with their own findings. Given the extent of these similarities, I believe this might constitute a form of plagiarism.

To illustrate, Figure 1 in the submission bears a strong resemblance to Figure 1 in the CFR paper. Furthermore, the layout and content of Table 1 (including the sequence of benchmarks), as well as Figures 4, 5, and Equations 3 and 4, are nearly identical to those in CFR.


[1] Calibrating Multi-modal Representations: A Pursuit of Group Robustness without Annotations, CVPR 2024 (https://openaccess.thecvf.com/content/CVPR2024/papers/You_Calibrating_Multi-modal_Representations_A_Pursuit_of_Group_Robustness_without_Annotations_CVPR_2024_paper.pdf)

---


1. The primary concern regarding this paper is that it appears to have been written with insufficient literature review.

The authors claim that previous research has not explored methods for instance-wise weighting, stating that existing approaches assign equal weights to groups. However, there are already methods that weight instances differently. For instance, the authors refer to LfF as a pseudo-labeling technique, yet LfF is not a pseudo-labeling method but rather an instance-weighting approach. This suggests a lack of understanding of the relevant literature.

- Lines 23-24 state, "this contrasts with existing group rebalancing weight strategies, which uniformly weigh all instances within a group." What distinguishes this work from existing methods that assign different weights to individual instances?

- Lines 90-91 state, "the relative importance of individual samples within each group remains an open question." I cannot agree with this assertion, as methods like LfF [1], DisEnt [2], and LWBC [3] already address this issue. The literature review appears to be lacking, especially given that LfF is even cited in the paper.

- Lines 284-285 mention, "such methods include pseudo-labeling with ERM model predictions (Nam et al., 2020)." However, Nam et al. does not represent pseudo-labeling. They utilize both the probabilities from the ERM model and the target model to calculate sample-specific weights. This indicates that the weights vary by instance and iteration. Is there confusion with JTT [4] here? Throughout the paper, it seems that the existence of LfF and the importance of instance weighting have been overlooked.


2. Methods that assign weights to instances should be compared with the proposed approach.

3. The requirement for group labels represents a significant limitation of this method. In this field, the challenges associated with obtaining group labels have already been recognized, leading to the development of approaches that learn without the need for group labels [1].

4. The overall composition of figures, tables, and equations in this paper bears striking similarities to the CFR [5] paper, yet the authors do not acknowledge this. They omit CFR while including their results and fail to cite it. The degree of similarity is too substantial to be coincidental. This lack of acknowledgment suggests potential plagiarism, particularly since Figure 1, Table 1, Figure 4, Figure 5, Equation 3, and Equation 4 mirror those in CFR. If the authors conducted their experiments in alignment with CFR, the omission of citation and comparative discussion raises ethical concerns. While one or two similar figures could be attributed to fair comparison, the extensive overlap across nearly all figures and tables is difficult to justify.

5. There is no comparison with Zhang & Ré, 2022 [6] despite the authors stating on lines 197-198, "Following (... Zhang & Ré, 2022,...), we employ..." Why have they not compared their results with this work? There appears to be no valid reason for this omission.


Overall, I believe the quality of this paper falls far short of top-conference standards.


[1] Learning from Failure: Training Debiased Classifier from Biased Classifier, NeurIPS 2020

[2] Learning Debiased Representation via Disentangled Feature Augmentation, NeurIPS 2021

[3] Learning debiased classifier with biased committee, NeurIPS 2022

[4] Just Train Twice: Improving Group Robustness without Training Group Information, ICML 2021

[5] Calibrating Multi-modal Representations: A Pursuit of Group Robustness without Annotations, CVPR 2024

[6] Contrastive Adapters for Foundation Model Group Robustness, NeurIPS 2022

**Questions:**

Please refer to the weaknesses.

---

### Note · Authors · 2024-11-13

I have read and agree with the venue's withdrawal policy on behalf of myself and my co-authors.